# Evolutionary Conflict Leads to Innovation: Symmetry Breaking in a Spatial Model of RNA-Like Replicators

**DOI:** 10.3390/life7040043

**Published:** 2017-11-02

**Authors:** Samuel H. A. von der Dunk, Enrico Sandro Colizzi, Paulien Hogeweg

**Affiliations:** Theoretical Biology and Bioinformatics, Utrecht University, 3584 CH Utrecht, The Netherlands; istidina@gmail.com

**Keywords:** RNA world, division of labour, symmetry breaking, catalysis, spatial model, higher-level selection, multilevel evolution

## Abstract

Molecules that replicate in trans are vulnerable to evolutionary extinction because they decrease the catalysis of replication to become more available as a template for replication. This problem can be alleviated with higher-level selection that clusters molecules of the same phenotype, favouring those groups that contain more catalysis. Here, we study a simple replicator model with implicit higher-level selection through space. We ask whether the functionality of such system can be enhanced when molecules reproduce through complementary replication, representing RNA-like replicators. For high diffusion, symmetry breaking between complementary strands occurs: one strand becomes a specialised catalyst and the other a specialised template. In ensemble, such replicators can modulate their catalytic activity depending on their environment, thereby mitigating the conflict between levels of selection. In addition, these replicators are more evolvable, facilitating survival in extreme conditions (i.e., for higher diffusion rates). Our model highlights that evolution with implicit higher-level selection—i.e., as a result of local interactions and spatial patterning—is very flexible. For different diffusion rates, different solutions to the selective conflict arise. Our results support an RNA World by showing that complementary replicators may have various ways to evolve more complexity.

## 1. Introduction

In present-day organisms, most functions are performed by specialised entities. At the molecular level, DNA is responsible for the storage of information, while a variety of proteins provides for most cellular functions [1]. Specialisation is apparent at every level of organisation in biology and multiple examples of division of labour have been the subject of research, e.g., DNA and proteins (e.g., [2,3,4]) or queen and worker bees [5].

Prebiotic entities most likely did not possess the same level of complexity [6]. According to the RNA World hypothesis [7,8], life started with self-replicating RNA molecules that could perform all functions needed for Darwinian evolution [9,10,11,12]. Such RNA replicases stored information and replicated one another. This raises an evolutionary problem. Because replicators cannot serve as template and catalyst at the same time, those that have a lower catalytic activity (i.e., that are more selfish) are favoured by selection [13,14,15,16]. Thus, evolution is expected to decrease catalytic activity until replicators go extinct.

Evolutionary stability of replicators can be achieved by incorporating a higher level of selection into a model [13]. Two classical approaches have been (1) to model replicators inside compartments (so-called protocells; [17,18,19,20,21]) or (2) to model replicators in space [19,22,23,24,25]. In the first case, protocells with more-catalytic internal replicators outcompete other protocells through faster accretion of resources. Cell-level dynamics provide explicit higher-level selection for increasing the catalytic activity of replicators, counteracting molecular-level selection for decreasing catalysis [21]. In the second case, higher-level selection is an emergent phenomenon that results from the spatial correlation between individuals. For limited diffusion, replicators with low catalytic activity are mostly surrounded by equally selfish individuals. In space, group-level dynamics provide implicit higher-level selection avoiding extinction [25].

Both modeling approaches achieve evolutionary stability of replicators when higher-level selection is sufficiently strong [21,25]. In compartment models, this is achieved by setting the cell volume *V* (the number of particles per cell), and in spatial models by setting the diffusion rate *D* (for a direct comparison, see [19]). For large *V* and *D*, higher-level selection prevents global extinction. There is an important difference between the two models due to a different implementation of higher-level selection (i.e., explicit versus implicit). Decreasing *V* in the compartment model intensifies cellular-level selection and, at some point, results in maximisation of catalysis. In contrast, decreasing *D* in the spatial model does not lead to maximisation of catalysis because group-level selection only emerges when catalytic rates are low (i.e., when replicators are close to extinction).

When the two levels of selection are comparable in magnitude, catalysis evolves to the bare minimum needed for survival [21,25]. At this selective balance, evolution is most creative, and complexity emerges. In simple models without complementary replication, replicators may self-organise into waves [25] or display evolutionary oscillations [21], pushing the upper limit of *D* and *V* for which they can survive. More recently, Takeuchi et al. [26] showed that, when compartments with complementary replicators are considered, symmetry breaks between the plus and the minus strand when the strength of the two selection pressures is similar. This allows complementary replicators to cope with even larger cell volumes (V=7500 versus V=2000). Here, we extend the insights from Takeuchi et al. [26] by studying complementary replicators in space.

The idea that symmetry breaking between complementary RNA strands might be an important step in prebiotic evolution comes from Szathmáry and Maynard Smith [27]. They postulated that specialisation of one strand as catalyst and the other as template could have alleviated the conflict between levels of selection. Such functional differentiation, resembling the division of labour between DNA and proteins, could indicate a gradual transition from selfish auto-replicators to cooperative genes and enzymes [28]. From a chemical perspective, functional symmetry appear to be rare in RNA sequence space, for example due to the GU wobble pair [29,30,31]. However, this does not make such sequences inaccessible to evolution. Therefore, we study emergent symmetry breaking in replicators without assuming functional differentiation a priori (cf. [24]), and without imposing structural/chemical constraints (cf. [32,33]). We also do not assume any trade-off or requirement for metabolic function (cf. [20,34]). Lastly, we do not impose higher-level selection (cf. [26]), but study the interplay of self-organisation, higher-level selection and the potential division of labour between complementary strands. We show that an evolutionary conflict can lead to innovation.

## 2. Results

### 2.1. The Model

The dynamics of RNA-like replicators were studied in a stochastic, individual-based and spatially-extended CA model based on CASH [25,35]. We used a two-dimensional, square lattice with a fixed number of sites, each of which can hold at most one replicator. The dynamics of replicators consist of replication, decay and diffusion. Complementary replication was implemented by assuming two types of molecules that serve as a template for each other’s formation: the plus and the minus strand (Figure 1). Because we do not consider chemical constraints, both types can in principle serve as a catalyst and as a template.

Replication happens in two steps. In the first step, two adjacent replicators, i.e., within the Moore neighbourhood, form a complex. In the second step, this complex makes a new replicator in an adjacent, empty grid cell—if available. The rate at which a replicator forms a complex with another replicator is given by one of four catalytic parameters depending on the identity of the catalyst and the template: $k_{\mathrm{ct}}$ where c∈{p,m} and t∈{p,m} (Figure 1). For instance, a plus strand can catalyse (i.e., form a complex with) a minus strand at the rate given by the plus strand’s value of $k_{\mathrm{pm}}$. We assume that most of the variation in the population exists in the difference between complementary strands; an individual catalyses all strands of one type (plus or minus) at the same rate [26]. Complexes span two sites of the lattice that diffuse together (see below). They can also break, as given by a fixed dissociation rate $k_{\mathrm{diss}}=0.25$. The growth rate of replicators depends on the number of complexes, which follows from their catalytic rates, and on available empty sites, which functions as resource ($k_{\mathrm{rep}}=1.0$). Each molecule has the same decay rate d=0.03. Diffusion is implemented as swapping of molecules, including complexes. We express the diffusion rate *D* as the average number of diffusion steps that molecules take in their lifetime: D=Δ/d giving D=3.3 for default parameters (Δ=0.1,d=0.03). For further details see Appendix B (“Details of the Model”).

To study the evolution of catalysis, we “mutated” the four catalytic parameters (Figure 1). New individuals receive the parameter values from their parental template. Mutations occur at the rate μ=0.005 per parameter ($\mu_{\mathrm{tot}}N\approx 4000$ for a 512 × 512 grid), and with a magnitude determined by a random draw from the uniform distribution $[-\delta_{\mu }/2$, $\delta_{\mu }/2]$, where $\delta_{\mu }=0.05$. They are bounded from above at $k_{\mathrm{ct}}=2.0$ (above this, mutations are reflected) but not from below. This prevents boundary effects close to zero such as the survival of replicators caused simply by reflected mutations. Nonetheless, rates below zero are treated as being zero in the model’s dynamics and whenever an average rate is calculated. To assess the effects of complementary replication, we study three simplified versions of the “full” model: the “basic” model from Colizzi and Hogeweg [25], a template model and a catalyst model. In the basic model, full symmetry (i.e., no distinction between plus and minus) is imposed by averaging the four parameters into a single catalytic parameter after a round of mutations: $k_{\mathrm{xx}}$. This way, the effective mutation rate is similar to the full model. In the template model, only template symmetry can break, allowing us to collapse four parameters into two: $k_{\mathrm{xp}}$ and $k_{\mathrm{xm}}$ (see Table 1). Here, catalytic activity does not depend on the identity of the catalyst. In the catalyst model, only catalytic symmetry can break, giving two parameters: $k_{\mathrm{px}}$ and $k_{\mathrm{mx}}$. In this case, catalytic activity does not depend on the identity of the template. Mutations in the template and catalyst model are normalised to the full model in the same manner as in the basic model.

### 2.2. Evolutionary Dynamics of the Model

An overview of the evolutionary dynamics of the model for different diffusion rates is shown in Figure 2. All the catalytic parameters are equal at the start. They initially decrease over evolutionary time, in line with previous studies [21,25,26]. This decrease in catalysis is accompanied by a decrease in population density and, for high diffusion rates, causes oscillations in the population density. In all cases, the symmetry in catalytic rates breaks and only two parameters remain positive.

#### 2.2.1. Evolutionary and Population Stability Depend on Diffusion

On average, replicators retain most catalysis when there is no diffusion, i.e., D=0. The population density is highest in this case. For slightly higher diffusion rates, the catalytic rates and population density in steady-state are lower (compare right panels Figure 2 for D=0 and D=10).

For D>20, increasing the diffusion rate decreases the stability of the evolutionary steady-state, as seen in oscillations of the global population density (Figure 2) and large empty patches in the field (Figure 2, snapshots). We observe a larger heterogeneity in catalytic rates, despite increased mixing, and we identify a subset of highly catalytic replicators (see snapshots).

When diffusion is too high (D>70), the system is globally evolutionarily unstable, as catalysis is decreased to the point where the entire population goes extinct.

#### 2.2.2. Diffusion Determines Interplay between Levels of Selection

We find two distinct ways in which multilevel selection can operate to yield macroscopic stability. With limited diffusion (D≤20), replicators with very low catalytic rates drive themselves to extinction locally, and are quickly replaced by neighbours with higher catalytic rates. Thus, group-level selection for cooperation balances with individual-level selection for selfishness at a small spatial scale and the system appears stable on the macroscale.

For higher diffusion rates (D>20), the selective forces are locally no longer in balance. Since group-level selection is relatively weaker and acting on a larger spatial scale than individual-level selection, catalysis is decreased until replicators locally go extinct. This produces empty space which, in sufficient quantities, elicits selection for invasion in replicators adjacent to the empty space. This way, emergent selection at the invasion- or wave-front can oppose selection for decreasing catalysis at the individual level. Hence, global evolutionary stability is achieved via large-scale oscillations in the population density (Figure 2).

The emergence of waves explains why populations that evolved under high diffusion display a higher diversity in evolutionary steady-state (Figure 2, snapshots; Figure 3). Namely, the wave-pattern causes spatial differences in the direction of evolution. Replicators increase catalysis at the wave-front and are exploited by replicators in the back, which in turn experience selection for decreasing catalysis. Thus, waves create two separate niches in the system that maintain catalysis together, i.e., through their coexistence.

In the long term, the differentiation between replicators at the front and the back of the wave can become too large to traverse with few mutations. Two sets of lineages form, each specialised for one niche (see e.g., D=40 in Figure 3). With higher diffusion rates, even higher diversity may be observed (Figure 3 and Appendix A).

#### 2.2.3. Diffusion Dictates Which Symmetry Breaks

Symmetry breaking of catalytic rates always occurs, but in different ways depending on diffusion. With four parameters, symmetry can break in three directions. The corresponding asymmetries are referred to as directional, template and catalytic (Table 1).

For D<40, directional asymmetry evolves (Table 1): $k_{\mathrm{pp}}$ and $k_{\mathrm{mm}}$ are decreased to zero whereas $k_{\mathrm{pm}}$ and $k_{\mathrm{mp}}$ remain positive. Both strand types can act as a catalyst but each can only catalyse the other strand type. Thus, each strand makes its own type. Because there is no functional differentiation between plus and minus strands, they are present in equal numbers in the population (right panel Figure 2).

For 40≤D≤70, catalytic and template asymmetry evolve (Table 1): $k_{\mathrm{mm}}$ and $k_{\mathrm{mp}}$ are decreased to zero, $k_{\mathrm{pp}}$ takes a small value and $k_{\mathrm{pm}}$ takes a large value. The functional differentiation between plus and minus strands (into specialised catalysts and templates, respectively) leads to an overwhelming majority of plus strands in the population (right panel Figure 2).

In almost all cases (except D=0), template symmetry breaks early during evolution. However, it is only maintained in the final equilibrium in combination with catalytic asymmetry (i.e., for 40≤D≤70). In the other cases (D<40), it is replaced by directional asymmetry.

#### 2.2.4. Complementary Replication Allows for Beneficial Evolutionary Dynamics

To further investigate the evolutionary consequences of symmetry breaking in the catalytic rates, the complementary replication model (the full model) was compared to the “basic” model from Colizzi and Hogeweg [25]. There is only one catalytic parameter in this model, so no asymmetry in catalytic rates can evolve. The parameters that evolved in the long term under different diffusion rates are shown in Figure 3. In both models, increasing the diffusion rate initially decreases the catalytic rate and at some point (D≥30) triggers the formation of waves (as seen in Figure 2), which leads to an increase in catalysis in a subset of replicators (Figure 3). The main difference is that the full model can survive at much higher diffusion rates than the basic model (D=70 versus D=30).

To understand the increased resistance of replicators to diffusion in the full model, we evolved replicators that could only break template symmetry (the template model) or catalytic symmetry (the catalyst model). The catalyst model can cope with marginally higher diffusion rates than the basic model (D=40) whereas the template model can cope with diffusion as high as the full model (in fact, up to D=80; Figure 3). Although replicators eventually have symmetric rates in the template model, template symmetry consistently breaks during evolution (Appendix A). Apparently, template asymmetry helps the system to self-organise so that it can cope with high diffusion.

#### 2.2.5. Causes and Consequences of Template Asymmetry

Template asymmetry leads to a large overrepresentation of the non-template strand (i.e., the plus), whether this is a better catalyst or not. This has two consequences. First, the detrimental effect of being a proficient catalyst is decreased, as molecules rarely engage in a catalytic interaction. In other words, evolving template asymmetry is an alternative for decreasing catalytic rates to be more available as a template. Hence, it is selected at the individual level. Second, replication is strongly limited by the number of usable templates. When empty space is available, a small increase in the number of usable templates can quickly increase the invasion rate. As there are many non-template strands, such increase can be achieved by using these plus strands slightly more as template (e.g., by increasing $k_{\mathrm{pp}}$ from zero to positive values, as seen in Figure 4). Thus, template symmetry breaking increases evolvability of the invasion rate (i.e., replication rate). As a result, areas of local extinction are more easily recolonised, preventing global extinction before the wave pattern self-organises.

#### 2.2.6. Catalytic and Directional Symmetry Breaking Enhance Replicators

The feature that sets apart directional and catalytic asymmetry from template asymmetry is that the first two are evolutionarily stable by themselves, while the latter only appears transiently (Figure 3). Directional and catalytic symmetry breaking do not extend the range of diffusion for which replicators survive, but they benefit replicators in a more subtle way. The full model generally maintains a larger population size in evolutionary steady-state than the template model (compare Figure 2 and Appendix A). Thus, directional and catalytic symmetry breaking can enhance replicators (for low and high diffusion rates, respectively). Below we investigate why catalytic asymmetry is evolutionarily stable. In these experiments we impose directional and template symmetry. We focus on non-evolving populations (i.e., no mutations) to show that catalytic symmetry breaking does not rely on emergent evolutionary properties of replicators, like increased evolvability.

We compared the growth curves of catalytically asymmetric and symmetric replicators (both have the same average rate, i.e., $k_{\mathrm{px}}=k_{\mathrm{mx}}=0.05$ or $k_{\mathrm{px}}=0.1$ and $k_{\mathrm{mx}}=0.0$) and observed differential growth of the first (Figure 5a). This reveals how asymmetric replicators can alleviate the conflict between selective forces. When selection for higher catalytic activity dominates (i.e., in a sparse environment), catalytic symmetry breaking enhances growth. Conversely, when selection for lower catalytic activity dominates (i.e., in a dense environment), catalytic symmetry breaking reduces growth, leaving replicators more available as template. In both cases, catalytically asymmetric replicators are selectively favoured over symmetric replicators. It should be noted that the growth curves in Figure 5a were obtained with extremely high diffusion. Differential growth of asymmetric replicators can still be observed with limited diffusion (i.e., in the range where replicators are evolutionarily stable). In that case growth curves cannot easily be constructed so we employed various competition experiments to determine which replicators grow faster in each environment. The details are provided in Appendix C (“Differential Growth with Limited Diffusion”).

#### 2.2.7. Differential Growth as a Result of Complex Population Dynamics

The strand composition that underlies differential growth can be seen in a propagating wave (Figure 5b, where the depth into the wave is a proxy for the local population density). Despite an overall majority of plus strands independent of the population density, fewer plus than minus strands are actually available where the population is dense (R≥6). Only at the wave-front (where the population is sparse), do free plus strands outnumber free minus strands. These observations show that when more catalysts (i.e., plus strands) are available, replicators grow faster. In contrast to template asymmetry, complex formation here is not limited by the availability of templates because all molecules can serve as template equally well. Instead, growth depends entirely on the number of available catalysts.

The overall majority of plus strands among replicators as a result of catalytic asymmetry can be explained by the inherent disadvantage of catalysts. Because plus strands spend time as catalyst in complex, they are less often available as template. Thus, more plus is produced from minus than vice versa. Even if complex formation is ignored, catalytic strands still have a disadvantage due to space (cf. [36]): catalysts tend to fill up their environment, reducing the amount of space available for their own offspring. Space adds to the majority of plus through a second mechanism (Figure 6). With limited diffusion, strands are likely to stay in the vicinity of the template from which they were produced. Since this parental template is per definition a complementary strand, plus strands will catalyse a minus strand more often than they will catalyse another plus strand, even if there were as many free plus as free minus strands in total.

Although the supply of catalysts is larger with catalytic asymmetry, the demand of plus strands is also larger. Because some plus strands still serve as template in a complex, not all plus strands can act as catalyst. This is not a problem when empty space is abundant, as complexes can quickly replicate, releasing constituent strands (catalyst and template). In a dense population however, replicators can get stuck in complex for some time. As a result, many plus strands are unavailable for replication—neither as template nor as catalyst—reducing the overall catalytic activity of the population in that case.

#### 2.2.8. Multiple Solutions to the Evolutionary Conflict

Catalytic asymmetry is always accompanied by template asymmetry in the full model evolved at high diffusion rates (40≤D≤70). Template asymmetry by itself only decreases growth, because the number of minus strands (i.e., templates) is decreased, inhibiting complex formation. The concomitant increase in plus strands does not benefit replication because plus strands are no better catalysts than minus strands in the template model. In the full model however, catalytic symmetry breaking enhances the functionality of the plus strand, thereby generating positive selection for increasing its abundance [26].

At low diffusion (D<40), the full model evolves directional asymmetry after an intermediate stage of template asymmetry. This suggests that directional symmetry breaking relies on the spatial correlation between a template and its offspring (see previous section). Due to this pattern—two adjacent strands are more likely to be complementary—biasing association rates towards the complement, improves a population’s catalytic potential. As only “catalytic pairs” (plus-minus and minus-plus) can get stuck in complex when the population is dense, more “non-catalytic pairs” (plus-plus and minus-minus) are left free (data not shown). Thus, directional symmetry breaking also yields differential growth: higher catalytic activity in a sparse environment and lower catalytic activity in a dense environment.

#### 2.2.9. Lineage Differentiation and Symmetry Breaking Together Prevent Extinction

Summarising the results so far, at high diffusion rates, the population differentiates in one or more subspecies with relatively high catalytic rates at the wave-front and selfish individuals with low catalytic rates in the back. This differentiation maintains the wave pattern. Symmetry breaking of the catalytic parameters benefits both subpopulations. At the wave-front, the higher catalytic potential can be exploited through the availability of empty space. In the denser back of the wave, the availability of minus strands gives asymmetric replicators a competitive advantage over symmetric replicators. Even when waves do not emerge, replicators benefit from symmetry breaking through this differential growth. Moreover, symmetry breaking can enhance evolvability of replicators, which helps to establish the wave pattern. Thus, complementary replication is exploited by replicators in multiple ways to adapt to self-organised environmental variability and prevent evolutionary extinction.

### 2.3. Two Competing Species: The Ecosystem-Based Solution for Maintaining Catalysis

So far, we have seen that complementary replication can enhance replicators belonging to the same species. Here we study the competition over evolutionary timescales of two different species that cannot cross-catalyse. We let the basic model and the catalyst model compete at a low diffusion rate (D=3.3), starting from equally low parameters ($k_{\mathrm{xx}}=0.1$ and $k_{\mathrm{px}}=k_{\mathrm{mx}}=0.1$, respectively). The species coexist through self-organisation into waves (Figure 7). Via differentiation of wave-front and back, the level of catalysis is enhanced in both species. We also observe symmetry breaking in the catalytic parameters at the wave-front of the catalyst model (Figure 7 and Figure 8).

The wave pattern is induced by the competition between species. No large empty patches emerge (cf. D>20, Figure 2), because each species maintains enough catalysis to survive. Nevertheless, the competition for space between species intensifies selection for higher catalytic activity at the border between the species. This allows catalytic replicators of one species to invade into the other species, forming a wave-front. Analogous to the case with a single species, selection at the wave-level causes differentiation of the wave-front and back (Figure 7 and Figure 8). This causes the increase of catalytic activity in the system, as a subset of replicators evolve to larger catalytic rates. As before (cf. D≥30, Figure 3), catalytic symmetry breaks at the wave-front, further enhancing the invasion rate of replicators. In short, interspecific competition triggers the formation of a highly functional, niche-rich ecosystem.

## 3. Discussion and Conclusions

### 3.1. Catalysis Is Maintained by a Diverse Ecosystem

We have shown that RNA-like replicators undergo functional differentiation to alleviate the evolutionary conflict caused by opposing levels of selection. This occurs in two ways: differentiation between lineages and differentiation within a lineage, i.e., symmetry breaking between the plus and the minus strand.

Replicators can differentiate into a high- and low-catalytic lineage though the self-organisation into waves, as in the model without complementary replication [25]. This kind of functional differentiation resulted from periodic disruptions, long replication times or high diffusion rates. Here, we have shown that competition between predefined species (i.e., with no cross-catalysis) also leads to separate lineages within a species. Therefore, this evolutionary mechanism is common and can be induced either by stronger individual-level selection (i.e., for long replication times and high diffusion rates) or by stronger group-level selection (i.e., for periodic disruptions or competing species). The differentiation of multiple lineages is also an effective solution to the evolutionary conflict. Species that have organised themselves into waves outperform those that lack this spatial pattern, in terms of catalysis.

Going beyond Colizzi and Hogeweg [25], we have shown that complementary replicators undergo lineage differentiation more easily, making them more resistant to high diffusion. They also maintain a higher diversity in the system (i.e., more than two lineages), adding to ecosystem complexity.

The other kind of functional differentiation, i.e., that between complementary strands, is observed in our full model for 40≤D≤70. The plus strand becomes a specialised catalyst while the other strand becomes a specialised template. This constitutes a *partial* division of labour: plus strands, despite being the only catalysts, still serve as a template occasionally to complete the replication cycle. The same differentiation is seen in the compartment model for large cell volumes [26]. In both cases, symmetry breaking relegates the evolutionary conflict from both molecular types to the plus strand only.

### 3.2. Population Dynamics as Substrate for Evolution

Through symmetry breaking, the catalytic activity of replicators is tuned by the number of catalysts and templates. Plus strands are overrepresented in complexes—serving both as catalyst and template whereas minus strands only serve as template. Thus, templates are always available in a population at steady-state. At the same time, asymmetric replicators make more catalysts in total. This can be exploited at the wave-front, because complexes are short-lived when there is much empty space. In this sense, complexes *modulate* the growth of asymmetric replicators depending on the local population density.

Via changes in the number of catalysts and templates, complementary replicators can increase or decrease their catalytic activity more rapidly on the evolutionary timescale too. This explains the increased resistance to high diffusion mentioned above.

A change in the ratio of catalysts over templates was also observed in two models without complex formation [24,34]. According to Könnyü and Czaran [34], catalytic asymmetry, resulting in many non-functional strands, “dilutes” the system in terms of functionality (see Discussion [34] (p. 9)). However, our results show that plus strands improve their catalytic activity considerably through evolution, thereby actually improving overall functionality of the system. Consequently, the survival range of the Metabolically Coupled Replicator System with functional differentiation may be underestimated [37].

In the model by Kim and Higgs [24], catalytic asymmetry in the polymerase leads to a majority of its functional strand, in line with our results (see e.g., [24] (Figure 7, p. 13)). They also assume catalytic asymmetry in the nucleotide synthetase, but there is no obvious reason for this: there are more non-functional than functional synthetase strands with catalytic asymmetry ([24] (Figure 7)). We expect this to be reversed when the synthesis of nucleotides takes time, because the functional strand will be less available for replication. Still, it is not clear whether catalytic asymmetry for functional RNAs other than replicases can be explained by population dynamics apart from potential chemical constraints. In the same spirit, we expect the polymerase to evolve a bias towards the non-functional synthetase strand, because such template asymmetry can increase the number of functional strands. In any case, complex ecosystems of RNA molecules may involve many more degrees of freedom than are commonly taken into account.

### 3.3. The Origins of Prebiotic Complexity

As explained in the introduction, previous studies have already indicated that evolutionary conflicts may be an important cause for innovation [26]. In the compartment model, evolutionary oscillations enable replicators to survive for larger cell volumes. In the spatial model, waves achieve the same effect with respect to diffusion.

Furthermore, symmetry breaking occurs in compartment and spatial models when selection pressures are comparable in strength. In the compartment model, symmetry breaking lowers the mutational pressure, thereby allowing replicators to “escape” molecular-level evolution for decreasing catalytic activity. Thus, symmetry breaking can make replicators more robust (some mutations have a smaller effect). We have shown that symmetry breaking can also make replicators more evolvable (some mutations have a larger effect). Nevertheless, changing the mutation rate or mutation step size has little effect on the outcome of evolution in the spatial model, unlike in the compartment model (data not shown). Earlier studies of RNA-like replicators with a complex genotype-to-phenotype map suggested a third explanation for functional differentiation of the plus and minus strand [32,33]: a lower rate of complex formation for two catalytic strands compared to a catalytic and a template strand (i.e., $k_{\mathrm{pp}}<k_{\mathrm{pm}}$) may inhibit folding of the catalytic strand onto itself. All these results align nicely with the findings from actual RNA landscapes (e.g., [31]). Thus, asymmetry between complementary strands is supported both by chemical constraints by and evolutionary dynamics.

Boza et al. [20] also show specialisation of plus and minus strands. However, catalytic asymmetry is assumed in their model and template symmetry breaks because of a predefined trade-off on the minus strand between metabolic activity and its ability to serve as template. When high metabolic activity is required, the minus strand evolves to be a worse template. The plus strand cannot metabolise, so the system can only meet the metabolic requirements by breaking template symmetry (i.e., minus becomes a worse template than plus).

In addition to the functional differentiation of the plus and the minus strand, we found another solution to the evolutionary conflict. For low diffusion, symmetry breaks such that both strands can only catalyse their complement (i.e., directional asymmetry). This solution was not observed in the compartment model as predefined higher-level dynamics leave evolution with fewer degrees of freedom. It is striking that space allows for additional outcomes of evolution. Nevertheless, in this case, the alternative catalytic mode may be less relevant, because the existence of such a chemical system seems unlikely.

Furthermore, we have shown that evolutionary stability of RNA-like replicators is not limited to unrealistically low diffusion rates. In fact, waves most likely allow replicators to cope with even more diffusion than we have shown here. The main challenge for surviving at high diffusion is the generation of the wave pattern, as explained for the template model. Waves that formed in the full model under D=70 remain evolutionarily stable when diffusion is increased to D=80 (data not shown) and probably beyond. In contrast, when replicators evolved under D=80 from the beginning, replicators could not form waves and went extinct (see Figure 2). This suggests that prebiotic evolution could have started from replicators bound to a surface but that, after spatial self-organisation, these replicators could have invaded harsher environments.

## Figures and Tables

**Figure 1 life-07-00043-f001:**
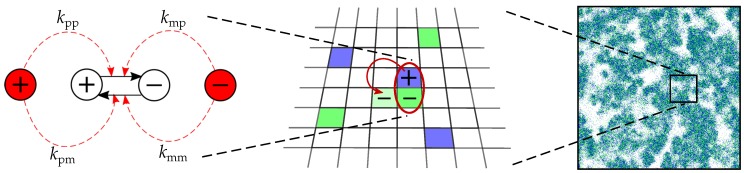
(**Left**) Schematic overview of the full model with the four catalytic parameters: $k_{\mathrm{pp}}$, $k_{\mathrm{mp}}$, $k_{\mathrm{pm}}$ and $k_{\mathrm{mm}}$. Solid arrows indicate replication (template to product); red dotted arrows indicate catalysis of a template by a catalyst (red); (**Middle**) Replication can happen when a complex has an empty site available in its neighbourhood; a complementary copy of the template is made; (**Right**) Replicators are modeled on a large, square grid with torroidal (wrapped) boundaries.

**Figure 2 life-07-00043-f002:**
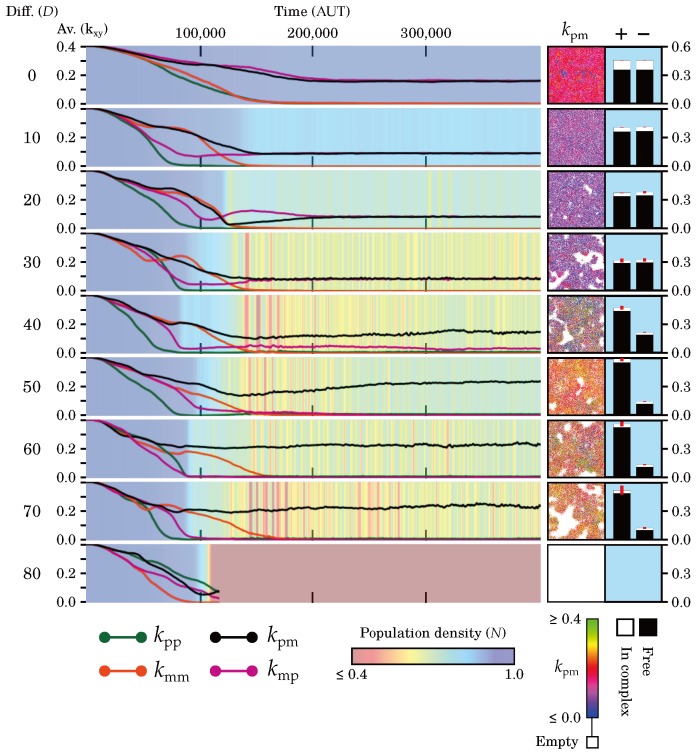
Evolutionary dynamics of the full model for different diffusion rates (see main text). The main plot shows the averages of the four parameters over time; background shows the corresponding population size. Two panels on the right provide further details of the population at the end of the timeplot—on the left, a snapshot of the field where individuals are colored according to their value of $k_{\mathrm{pm}}$; on the right, a barplot that shows the number of plus and minus strands that are free and in complex (red bars span one standard deviation above and below the mean of five independent time samples per simulation). All simulations were performed on a 512 × 512 grid and with default parameters. For D=10 and D=70, videos are available in the Appendix A. For the steady-state distributions of the catalytic rates, see Figure 3.

**Figure 3 life-07-00043-f003:**
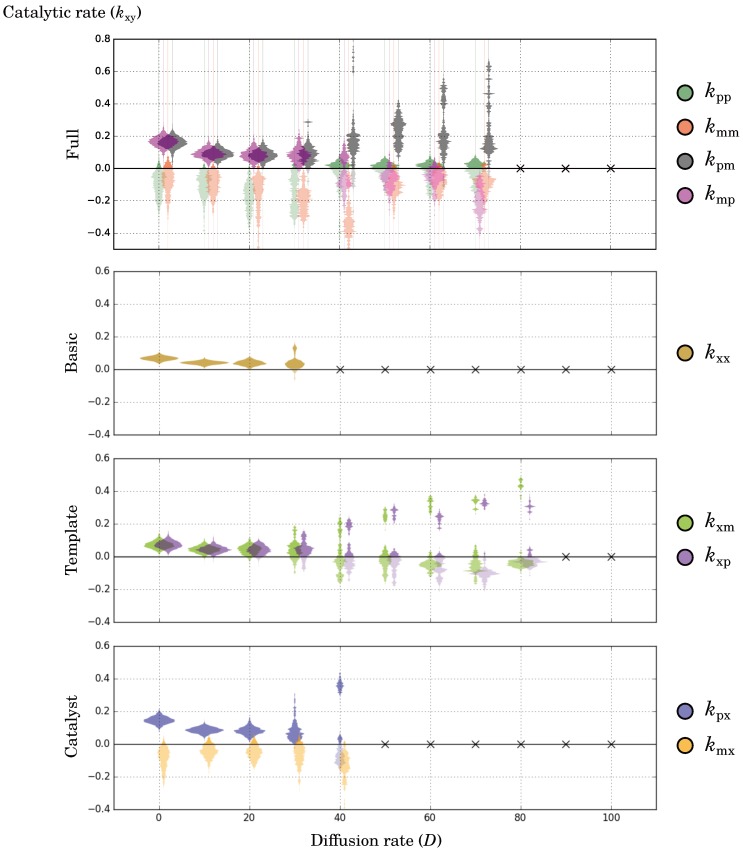
Distribution of the evolved catalytic rates for different diffusion rates, per model. Samples were taken once the population distribution remained similar for long periods of time; nevertheless, rates may drift up and down considerably in the cases where waves formed. A small cross is marked on the line y=0 when a species went extinct. All simulations were performed on a 512 × 512 grid and with default parameters. Because negative rates are treated as zero, they were given a lighter shading.

**Figure 4 life-07-00043-f004:**
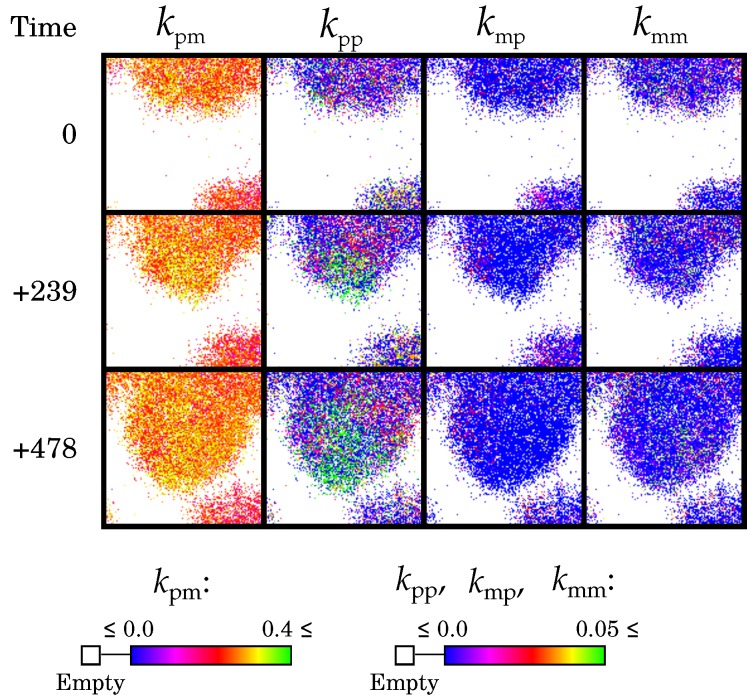
The emergence of a wave-front in the full model: replicators can adapt quickly by changing $k_{\mathrm{pp}}$. Figure shows the same 128 × 128 window over time. The wave emerged for D=60 at approximately 180,000 AUT (see Figure 2).

**Figure 5 life-07-00043-f005:**
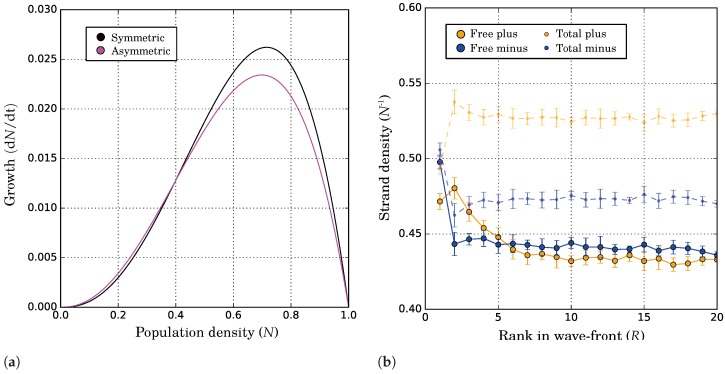
Differential growth underlies catalytic symmetry breaking. (**a**) Growth curves (best fits to data) of catalytically asymmetric and symmetric replicators under high diffusion (D=670) and with no decay (d=0). The curves were measured in separate simulations, each on a 512 × 512 grid; (**b**) Composition of a propagating wave of asymmetric replicators under low diffusion (D=3.3). Relative strand densities (normalised for total population density) are shown as a function of the horizontal distance to the wave-front (where position R=1 is the foremost replicator per row). Error bars give one standard deviation above and below the mean for five replicate runs, each on a 200 × 10,000 grid (length × height). In both (**a**,**b**), $k_{\mathrm{px}}=0.1$, $k_{\mathrm{mx}}=0$ for catalytically asymmetric replicators; in (**a**), $k_{\mathrm{px}}=k_{\mathrm{mx}}=0.05$ for symmetric replicators.

**Figure 6 life-07-00043-f006:**
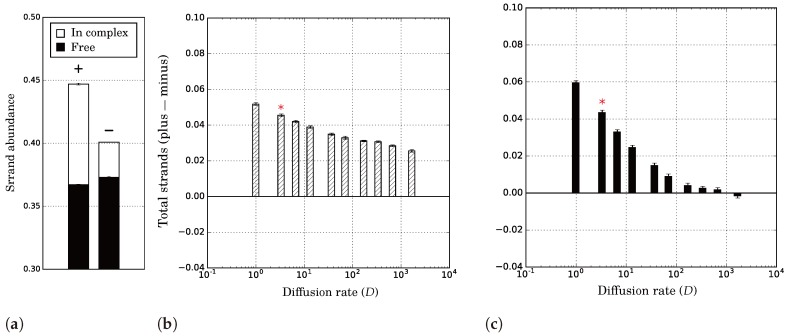
Local interactions enhance asymmetry in the number of plus and minus strands, given catalytic asymmetry in the association rates (i.e., $k_{\mathrm{px}}=0.1$, $k_{\mathrm{mx}}=0$). (**a**) Relative strand densities for limited diffusion (D=3.3); (**b**) Increasing the diffusion rates decreases the overall majority of plus strands; (**c**) With limited diffusion, strands are more likely to be close to a complementary strand than a equivalent strand. Error bars give one standard deviation of 25 independent samples from a single population in ecological equilibrium. Red asterisks in (**b**,**c**) signify the diffusion rate that corresponds to (**a**). Populations were simulated on a 512 × 512 grid without mutations (μ=0), and otherwise default parameters.

**Figure 7 life-07-00043-f007:**
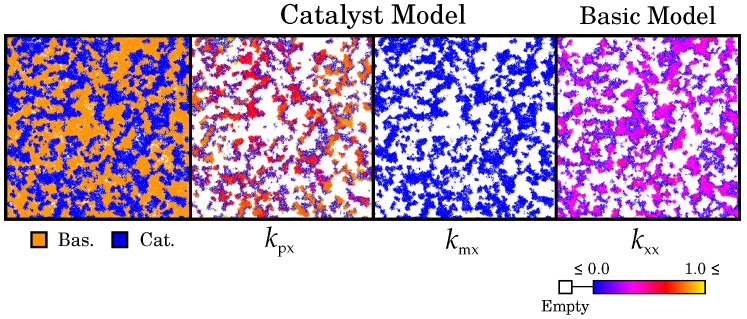
Spatial dynamics of the competition experiment at evolutionary steady-state. Coexistence of the two species is mediated by waves. The establishment of the wave-pattern is shown in Appendix A (with a lower color resolution).

**Figure 8 life-07-00043-f008:**
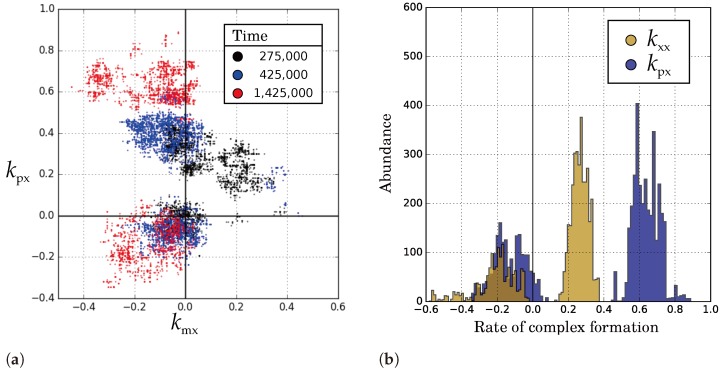
Both species have differentiated in two lineages at T=1,425,000. (**a**) In the catalyst model, $k_{\mathrm{mx}}$ is decreased and $k_{\mathrm{px}}$ splits into a lineage with high values at the wave-front and a lineage with low values at the wave-back; (**b**) Comparison of $k_{\mathrm{px}}$ in the catalyst model and $k_{\mathrm{xx}}$ in the basic model shows that the average catalytic rate at the wave-front is similar between the species ($\frac{k_{\mathrm{px}}+k_{\mathrm{mx}}}{2}\approx k_{\mathrm{xx}}$).

**Table 1 life-07-00043-t001:** Symmetry can break in three different ways in the model. In the case of catalytic and template asymmetry, we will refer to the more catalytic strand and the better template strand as plus and minus, respectively. Note that the double-strand replicators could evolve such that one strand is the better catalyst and the better template, but this is never observed. Directional asymmetry could also break in the opposite way (i.e., both strands preferentially catalysing the same strand type), but this does not happen.

Asymmetry	Description	Parameters	Observed
Directional	Both strands preferentially catalyse their complement.	$k_{\mathrm{pm}},k_{\mathrm{mp}}$$>k_{\mathrm{pp}},k_{\mathrm{mm}}$	equilibrium (D<40)
Template	One of the strands (minus) is a better template.	$k_{\mathrm{mm}},k_{\mathrm{pm}}$$>k_{\mathrm{pp}},k_{\mathrm{mp}}$	transient (10≤D≤70),
			equilibrium (40≤D≤70)
Catalytic	One of the strands (plus) is a better catalyst.	$k_{\mathrm{pp}},k_{\mathrm{pm}}$$>k_{\mathrm{mm}},k_{\mathrm{mp}}$	equilibrium (40≤D≤70)

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
