# Peer review of "Evolutionary Conflict Leads to Innovation: Symmetry Breaking in a Spatial Model of RNA-Like Replicators"

_life, 2017, doi:10.3390/life7040043_

Round 1

Reviewer 1 Report

Fig 2 is complex and takes a long time to digest. After some time, I think I am appreciating all the bits of it. However, there is still something that is not easy to see. Whereas at low D it is clear that k­pm and kmp remain non-zero, at intermediate D (50 or 60) it looks as though only kpm is non zero. In fact kpp is also non-zero but very small. Is there some way to make this clearer in the figure. Without this, the reader cannot understand why the system survives. A system with only kpm non-zero would die. One possibility is to plot the density of strands with kct > 0 for each ct combination, instead of plotting the mean of kct. There must be large numbers of strands with kpp > 0 in the above case.

An interesting observation in this paper is the 'cyclic phase' where P replicates M and M replicates P (i.e. kpm and kmp are non-zero). This is more or less like a hypercycle model, although there are some differences because P and M are copied from one another and are therefore not independent. Can something be said about the relationship of this model to a model of a hypercycle with two components? In the paper on strand symmetry breaking in protocells (Takeuchi et al 2017)  there doesn't seem to be any mention of a cyclic phase. Is there a reason why a cyclic phase cannot form in the protocell model, or was it simply not looked for?

There is a brief mention on line 65 that functional asymmetry may be common due to GU pairs. In fact, I would assume that it is overwhelmingly common, and furthermore, it is almost impossible to find a sequence that is functional in both strands. I think there should be some clear statement that two complementary RNA sequences almost always fold to completely different secondary structures and are therefore unlikely to both have a function. My main criticism of this paper is that the inherent rarity of sequences that are functional as both P and M is glossed over. Hence the starting point of the simulations in which all four k's are positive seems highly unlikely from an origin of life point of view.

            For the authors' benefit, I will note current work in my own group. We are using RNA folding algorithms to measure the frequency of sequences that have the same structure in both strands. For each target structure considered, we find that the frequency of these dual-folders is always much less than the frequency of sequences that fold to the correct structure in only one strand, but there are at least some secondary structures for which we do observe at least a small number of dual-folders. We then consider how likely it is that a dual-folder will replicate to a sequence that folds correctly in both strands, or in one strand, or neither. As there are more constraints on dual folders, the likelihood of deleterious mutations is higher. Hence the error threshold is lower than for a sequence with function in only one strand. Another aspect that is glossed over in this paper is the details of the mutation process - it is simply assumed that all four of the k's can mutate separately. The conclusion from our work in progress is that the dual folders can only beat the single folders when the error rate is low, and that the dual folders can only exist in the first place for some unusual secondary structures where the frequency of dual folders in sequence space is not too small.

I am not sure of the logic of how a catalyst discriminates between P and M strands - e.g. what property of the catalyst sequence makes kpm different from kpp? Is it binding to some sequence specific part of the template, or to some general feature that is independent of sequence? The binding is tacitly assumed to be a property of the catalyst, not the template. This means that if kpp is large it binds to all P templates rapidly. In the present model it is not possible for a template to evolve that binds to all catalysts faster. At the risk of using anthropomorphic language - for whose benefit does the template asymmetry evolve? It would seem worthwhile in future considering a model where the ability to be a template is a property of the template sequence itself (or indeed of both template and catalyst).

From the point of view of origin of life and early evolution of complexity, it seems that what we are looking for is a general replicase, not a sequence-specific one. Once we have a general replicase we can add all sorts of other ribozymes that have different functions, which are all copied by the same replicase. The cyclic phase in this paper, where one specific P replicates its own M and vice versa, seems very specific - an evolutionary dead end. How can this same replicase be used to replicate sequences with different functions and structures.

This model has the binding of the catalyst to the template as a separate step from the replication. This means that the catalyst spends a significant time where it cannot be a template. Several of the arguments about symmetry breaking in this paper depend crucially on this. For example, if a catalyst no longer binds to one type of strand, this gives it an automatic advantage as a template. Please can you add a discussion paragraph where you highlight which results depend on this and which would still be true in a model where replication is treated as a single step?

            In the present model, symmetry breaking is selected for (mutation plays little role). In the current work I mentioned above, mutation is key - symmetry breaking is a loss of function, like an error threshold. So you can lose function in the minus strand due to error accumulation, and then you can select for faster replication of the minus template (kpm > kpp) even if replication is treated as a single step.

Author Response

p { margin-bottom: 0.1in; line-height: 120%; }

Fig 2 is complex and takes a long time to digest. After some time, I think I am appreciating all the bits of it. However, there is still something that is not easy to see. Whereas at low D it is clear thatkpmand kmp remain non-zero, at intermediate D (50 or 60) it looks as though only kpm is non zero. In fact kpp is also non-zero but very small. Is there some way to make this clearer in the figure. Without this, the reader cannot understand why the system survives. A system with only kpm non-zero would die. One possibility is to plot the density of strands with kct > 0 for each ct combination, instead of plotting the mean of kct. There must be large numbers of strands with kpp > 0 in the above case.

Figure 2 is indeed complex, which is why we chose to only plot the mean catalytic rates instead of their distribution. The distributions can however be seen in Figure 3, revealing that most replicators have positive values of kpp for higher values of D. We added a reference to Figure 3 (legend of Figure 2).

An interesting observation in this paper is the 'cyclic phase' where P replicates M and M replicates P (i.e. kpm and kmp are non-zero). This is more or less like a hypercycle model, although there are some differences because P and M are copied from one another and are therefore not independent. Can something be said about the relationship of this model to a model of a hypercycle with two components? In the paper on strand symmetry breaking in protocells (Takeuchi et al 2017) there doesn't seem to be any mention of a cyclic phase. Is there a reason why a cyclic phase cannot form in the protocell model, or was it simply not looked for?

The 'cyclic phase' is indeed a new outcome of evolution that we found with respect to the compartment model (Takeuchi et al., 2017). Because this mode fully depends on a microscopic spatial pattern (strands are often still close to their 'template parent'), it does not evolve in a compartment model where replicators within each compartment are well-mixed. Moreover, symmetry breaking does not occur in the compartment model when vesicles are too small. We added a remark in the Discussion (“The Origins of Prebiotic Complexity”, lines 373-377).

There is a brief mention on line 65 that functional asymmetry may be common due to GU pairs. In fact, I would assume that it is overwhelmingly common, and furthermore, it is almost impossible to find a sequence that is functional in both strands. I think there should be some clear statement that two complementary RNA sequences almost always fold to completely different secondary structures and are therefore unlikely to both have a function. My main criticism of this paper is that the inherent rarity of sequences that are functional as both P and M is glossed over. Hence the starting point of the simulations in which all four k's are positive seems highly unlikely from an origin of life point of view.

We rephrased this part of the introduction (Introduction, lines 64-66). We agree that asymmetry is most likely far more common. However, in this study we wish to see if we can explain symmetry breaking purely from the (evolutionary) dynamics of replicators, taking an “agnostic” stance towards the actual RNA sequence space. As for the initialisation of the full model, the outcome of evolution is independent of the initial parameter values (as long as they are not too low); starting with symmetric values shows most clearly that symmetry breaks.

For the authors' benefit, I will note current work in my own group. We are using RNA folding algorithms to measure the frequency of sequences that have the same structure in both strands. For each target structure considered, we find that the frequency of these dual-folders is always much less than the frequency of sequences that fold to the correct structure in only one strand, but there are at least some secondary structures for which we do observe at least a small number of dual-folders. We then consider how likely it is that a dual-folder will replicate to a sequence that folds correctly in both strands, or in one strand, or neither. As there are more constraints on dual folders, the likelihood of deleterious mutations is higher. Hence the error threshold is lower than for a sequence with function in only one strand. Another aspect that is glossed over in this paper is the details of the mutation process - it is simply assumed that all four of the k's can mutate separately. The conclusion from our work in progress is that the dual folders can only beat the single folders when the error rate is low, and that the dual folders can only exist in the first place for some unusual secondary structures where the frequency of dual folders in sequence space is not too small.

These are interesting results which appear to coincide with our findings that prebiotic replicators would most likely have been asymmetric. However, in our present study we worked without realistic chemical constraints.

I am not sure of the logic of how a catalyst discriminates between P and M strands - e.g. what property of the catalyst sequence makes kpm different from kpp? Is it binding to some sequence specific part of the template, or to some general feature that is independent of sequence? The binding is tacitly assumed to be a property of the catalyst, not the template. This means that if kpp is large it binds to all P templates rapidly. In the present model it is not possible for a template to evolve that binds to all catalysts faster. At the risk of using anthropomorphic language - for whose benefit does the template asymmetry evolve? It would seem worthwhile in future considering a model where the ability to be a template is a property of the template sequence itself (or indeed of both template and catalyst).

We have added the rationale for this to the model description (Results, lines 86-88). It is indeed a simplification of the model, which allows us to discriminate between plus and minus strands. Given that most of the variation in the population is due to the difference between complementary strands, this assumption would be justified.

From the point of view of origin of life and early evolution of complexity, it seems that what we are looking for is a general replicase, not a sequence-specific one. Once we have a general replicase we can add all sorts of other ribozymes that have different functions, which are all copied by the same replicase. The cyclic phase in this paper, where one specific P replicates its own M and vice versa, seems very specific - an evolutionary dead end. How can this same replicase be used to replicate sequences with different functions and structures.

We also doubt that this is a real solution for RNA (see again Discussion, lines 373-377). Nonetheless, it is an interesting outcome of our model that underscores the flexibility of evolution in a spatial system. We think the partial division of labour arising at high diffusion is more interesting for the RNA World. Not only does this seem chemically feasible, but it is also found in the compartment model (Takeuchi et al., 2017) and the RNA model with folding structures (Colizzi & Hogeweg, 2014).

This model has the binding of the catalyst to the template as a separate step from the replication. This means that the catalyst spends a significant time where it cannot be a template. Several of the arguments about symmetry breaking in this paper depend crucially on this. For example, if a catalyst no longer binds to one type of strand, this gives it an automatic advantage as a template. Please can you add a discussion paragraph where you highlight which results depend on this and which would still be true in a model where replication is treated as a single step?

In the Results (“Differential Growth as a Result of Complex Population Dynamics”, line 234-236), we address this: because of local interactions, the results will most likely be the same as catalysts have an disadvantage in space.

In the present model, symmetry breaking is selected for (mutation plays little role). In the current work I mentioned above, mutation is key - symmetry breaking is a loss of function, like an error threshold. So you can lose function in the minus strand due to error accumulation, and then you can select for faster replication of the minus template (kpm > kpp) even if replication is treated as a single step.

This is an interesting potential other mechanism for symmetry breaking. In contrast to the compartment model, our results are largely independent of mutation rates. We have added this point to Appendix A (lines 410-413) and the Discussion (lines 357-359).

Reviewer 2 Report

General comments:

Overall, I find this manuscript original and its content interesting and significant to the community. The points raised by the authors and the discussion are both generally accurate and relevant. 

However, I would like to address some issues:

My main concern on this work is the lack of an analytical framework. Several points can be made in regards to this:

- Most choices of parameters values do not seem to be justified or supported by a theoretical background. For example, the dissociation rate k_diss is fixed, but no discussion of what is the impact of this value is provided, same for mutation rates and subsequent distributions.

- A remarkable result concerns the appearance of a characteristic diffusion rate beyond which higher-level selection takes over and dominates the system's dynamics particularly playing a role in inducing symmetry breaking. However, the lack of a theoretical supporting argument makes it difficult to understand what dynamical processes are producing this effect and why this happens (sharply?) at a given characteristic point. Numerical values for D in this case can seem arbitrary when no such framework is provided.

Hence, in the interest of strengthening the overall message, I would like to suggest the introduction of an analytical model, which could provide a more cohesive view. A simplified null model (not necessarily involving the whole complexity of the in silico simulations) could work and serve as a backbone connecting all the different scenarios discussed in the manuscript. Maybe focusing on each broken symmetry (Table 1), and trying to study the system on these three limiting cases could shed some light on the underlying dynamics of this emergent phenomena.

-------------------

Other minor issues:

- Could you explain why data from two distinct simulations is given in figure 2 (D=50), figure 3 and video S2? Is there a reason why the three cannot be extracted from the same simulation?

- In lines 320-321 reference to similar replicator dynamics is mentioned, but no references are given. Could you provide a reference for quorum sensing in the manuscript that relates to what is being exposed in this corresponding paragraph?

- Figure 8b seems not to have compiled properly.

Minor corrections:

Lines 42-43 and 56-57: here the authors use V, which first is said to indicate "volume" and then number of particles in a cell (26). This can create confusion, please add a line defining the volume as accordingly or ommit explicit numerical values.

Minor corrections at the References:

- Ref 5: lacks capital letters in editorial: "Hardvard University Press" (not "harvard university press).

- Refs 12, 23, 26 & 11: lack the final article's final page.

Author Response

p { margin-bottom: 0.1in; line-height: 120%; }

Overall, I find this manuscript original and its content interesting and significant to the community. The points raised by the authors and the discussion are both generally accurate and relevant.

However, I would like to address some issues:

My main concern on this work is the lack of an analytical framework. Several points can be made in regards to this:

- Most choices of parameters values do not seem to be justified or supported by a theoretical background. For example, the dissociation rate k_diss is fixed, but no discussion of what is the impact of this value is provided, same for mutation rates and subsequent distributions.

We added a model section to the appendix where we explain parameter choices and provide reaction equations and ODE's to clarify model details (Appendix A). Note, however, that individual spatial embedding is crucial for our results and prevents analytical treatment. In the ODE model, replicators would evolve into extinction.

- A remarkable result concerns the appearance of a characteristic diffusion rate beyond which higher-level selection takes over and dominates the system's dynamics particularly playing a role in inducing symmetry breaking. However, the lack of a theoretical supporting argument makes it difficult to understand what dynamical processes are producing this effect and why this happens (sharply?) at a given characteristic point. Numerical values for D in this case can seem arbitrary when no such framework is provided.

Hence, in the interest of strengthening the overall message, I would like to suggest the introduction of an analytical model, which could provide a more cohesive view. A simplified null model (not necessarily involving the whole complexity of the in silico simulations) could work and serve as a backbone connecting all the different scenarios discussed in the manuscript. Maybe focusing on each broken symmetry (Table 1), and trying to study the system on these three limiting cases could shed some light on the underlying dynamics of this emergent phenomena.

We have chosen to explain D in a natural way: it stands for the average number of diffusion steps a replicator makes during its lifetime. We then study the qualitative behaviour of the model along the D-axis, but we also compare the full model with simplified versions, showing that quantitatively, the complementary replication model has extended its survival range.

Other minor issues:

- Could you explain why data from two distinct simulations is given in figure 2 (D=50), figure 3 and video S2? Is there a reason why the three cannot be extracted from the same simulation?

We changed this (see legend Figure 3). For Figure S2 this is not important, so we removed the confusing remark.

- In lines 320-321 reference to similar replicator dynamics is mentioned, but no references are given. Could you provide a reference for quorum sensingin the manuscript that relates to what is being exposed in this corresponding paragraph?

This reference was slightly out of line, so we removed it.

- Figure 8b seems not to have compiled properly.

We cannot find a problem..?

Minor corrections:

Lines 42-43 and 56-57: here the authors use V, which first is said to indicate "volume" and then number of particles in a cell (26). This can create confusion, please add a line defining the volume as accordingly or ommit explicit numerical values.

Done.

Minor corrections at the References:

- Ref 5: lacks capital letters in editorial: "Hardvard University Press" (not "harvard university press).

- Refs 12, 23, 26 & 11: lack the final article's final page.

Done.

Reviewer 3 Report

The manuscript by Prof. Hogeweg and co-workers investigates how division of labour can evolve in complementary strands of RNA-like replicators. An initially similar complementary pair of RNA replicators evolve their catalytic activity toward the replication of their own strand and their complementary strand. The prebiotic dynamical system is set on surfaces, which was an important stage in the origin of life before the emergence of the first cells. The paper shows that division of labour between the strands can evolve, as one strand will be mostly catalytic and the other mostly a template. They also report a two species investigation in which 2 pairs of RNA replicators coexist in the spatially explicit setting. This last result plays a prominent role in the discussion albeit only mentioned briefly in the results. I suggest lengthening that part a bit.

There are two parts of the manuscript that could be improved: the introduction and the method section.

In the introduction, there are some further references that you might consider adding. When referencing the RNA world, I suggest adding the book of Yarus (Yarus, M. Life from an RNA world: The ancestor within. Harvard University Press: Harvard, USA, 2011.) as a more recent reference. Surface bound RNA-like replicators are extensively studied by Tamás Czárán and Balázs Könnyű. Their recent review (Czárán, T.; Könnyű, B.; Szathmáry, E. Metabolically coupled replicator systems: Overview of an RNA-world model concept of prebiotic evolution on mineral surfaces. J. Theor. Biol. 2015, 381, 39–54.) is a good overview of their work.

The method section refers to prior work for detailed methods. I think the description of the way the complex formation is described at the moment is not sufficient. This is the key element, as those pairs of molecules in complex are not available for replication. The result section elaborates its importance (and it is a fully justifiable assumption), but it is something one needs to deduce from the results and not from the methods. Please describe how do you record complex. Do you join them into one entity or somehow record that there is a link between certain cells?

One of our previous paper which was also dealing with division of labour between RNA replicators was mentioned in the paper. I agree that study presented in this manuscript is more general in the sense, that all possible catalytic rates are evolvable, while we have set some of them to zero. On the other hand, while the two strands can both be an enzyme, I find it less plausible that both can be a replicase ribozyme. Replicase ribozymes are not easy to find, actually there is no such general enzyme yet, despite nearly two decades of search for it.

In summary, this manuscript is a well written and will be an important addition to the literature of the theoretical foundation of origins of life research. I strongly recommend it to be accepted.

Minor comments:

P2L40 I think “to avoid extinction” is more appropriate.

Variables should be italicized. But subscripts that are not variables should be normal text, like rep, diss, mm, pp, pm and mp which names comes from minus and plus. (kmm)

In the figures k should be italicized. Furthermore, usually fonts in figures are of sans-serif type (e.g. Arial) and not serif type (like Times).

Fig 2: In “Diff (D)” D should be italicized

Fig 5. The X and the Y axes label are of different sizes. The N in the labels should be italicized.

Fig. 6. The X and the Y axes label are of different sizes. In the (b) and (c) panels the part between -0.04 – -0.02 and 0.08 – 0.10 are not necessary. That way the columns can be better seen. You should indicate that the X scale is logarithmic.

The acknowledgement part is from a template and not actual acknowledgement. Please rewrite.

References should be made more uniform in the sense of journal names should be italicized, capitalized and abbreviated.

Ref [1] and [21] vol. and p. should be written.

Ref [7], [26] need pages or article number

Ref [8] book title should be italicized

Ref [27] The roman number II should not be written as ii

Author Response

p { margin-bottom: 0.1in; line-height: 120%; }

The manuscript by Prof. Hogeweg and co-workers investigates how division of labour can evolve in complementary strands of RNA-like replicators. An initially similar complementary pair of RNA replicators evolve their catalytic activity toward the replication of their own strand and their complementary strand. The prebiotic dynamical system is set on surfaces, which was an important stage in the origin of life before the emergence of the first cells. The paper shows that division of labour between the strands can evolve, as one strand will be mostly catalytic and the other mostly a template. They also report a two species investigation in which 2 pairs of RNA replicators coexist in the spatially explicit setting. This last result plays a prominent role in the discussion albeit only mentioned briefly in the results. I suggest lengthening that part a bit.

We agree that the two-species system is an interesting system. However, this system is quite different from the rest of the story, so we chose to focus only on those parts that can be connected to the functional differentiation that we observed in the systems with single species (i.e. differentiation through wave formation and functional differentiation of plus and minus strands).

There are two parts of the manuscript that could be improved: the introduction and the method section.

In the introduction, there are some further references that you might consider adding. When referencing the RNA world, I suggest adding the book of Yarus (Yarus, M. Life from an RNA world: The ancestor within. Harvard University Press: Harvard, USA, 2011.) as a more recent reference. Surface bound RNA-like replicators are extensively studied by Tamás Czárán and Balázs Könnyű. Their recent review (Czárán, T.; Könnyű, B.; Szathmáry, E. Metabolically coupled replicator systems: Overview of an RNA-world model concept of prebiotic evolution on mineral surfaces. J. Theor. Biol. 2015, 381, 39–54.) is a good overview of their work.

Thank you; we added a reference to Czaran et al. (2015) in the Discussion (line 335). Unfortunately, we do not have the book by Yarus that you mention.

The method section refers to prior work for detailed methods. I think the description of the way the complex formation is described at the moment is not sufficient. This is the key element, as those pairs of molecules in complex are not available for replication. The result section elaborates its importance (and it is a fully justifiable assumption), but it is something one needs to deduce from the results and not from the methods. Please describe how do you record complex. Do you join them into one entity or somehow record that there is a link between certain cells?

We added an appendix with more technical details of the model to the manuscript.

One of our previous paper which was also dealing with division of labour between RNA replicators was mentioned in the paper. I agree that study presented in this manuscript is more general in the sense, that all possible catalytic rates are evolvable, while we have set some of them to zero. On the other hand, while the two strands can both be an enzyme, I find it less plausible that both can be a replicase ribozyme. Replicase ribozymes are not easy to find, actually there is no such general enzyme yet, despite nearly two decades of search for it.

We agree, yet our aim was to “ignore” RNA chemistry and see if simple (unconstrained) complementary replicators would break symmetry. Both replicator dynamics and RNA landscapes seem to point at functionally asymmetric RNA replicases. This point is also added to the Discussion (“The Origins of Prebiotic Complexity”, lines 362-364).

In summary, this manuscript is a well written and will be an important addition to the literature of the theoretical foundation of origins of life research. I strongly recommend it to be accepted.

Minor comments:

P2L40 I think “to avoid extinction” is more appropriate.

Variables should be italicized. But subscripts that are not variables should be normal text, like rep, diss, mm, pp, pm and mp which names comes from minus and plus. (kmm)

In the figures k should be italicized. Furthermore, usually fonts in figures are of sans-serif type (e.g. Arial) and not serif type (like Times).

Fig 2: In “Diff (D)” D should be italicized

Fig 5. The X and the Y axes label are of different sizes. The Nin the labels should be italicized.

Fig. 6. The X and the Y axes label are of different sizes. In the (b) and (c) panels the part between -0.04 – -0.02 and 0.08 – 0.10 are not necessary. That way the columns can be better seen. You should indicate that the X scale is logarithmic.

The acknowledgement part is from a template and not actual acknowledgement. Please rewrite.

References should be made more uniform in the sense of journal names should be italicized, capitalized and abbreviated.

Ref [1] and [21] vol. and p. should be written.

Ref [7], [26] need pages or article number

Ref [8] book title should be italicized

Ref [27] The roman number II should not be written as ii

We have adapted most of the above in our manuscript.
